# Dexterous Non-Prehensile Manipulation for Ungraspable Objects via Extrinsic Dexterity

## Abstract

Objects with large base areas become ungraspable when they exceed the end-effector's maximum aperture. Existing approaches address this limitation through extrinsic dexterity, which exploits environmental features for non-prehensile manipulation. While grippers have shown some success in this domain, dexterous hands offer superior flexibility and manipulation capabilities that enable richer environmental interactions, though they present greater control challenges. Here we present ExDex, a dexterous arm-hand system that leverages reinforcement learning to enable non-prehensile manipulation for grasping ungraspable objects. Our system learns two strategic manipulation sequences: relocating objects from table centers to edges for direct grasping, or to walls where extrinsic dexterity enables grasping through environmental interaction. We validate our approach through extensive experiments with dozens of diverse household objects, demonstrating both superior performance and generalization capabilities with novel objects. Furthermore, we successfully transfer the learned policies from simulation to a real-world robot system without additional training, further demonstrating its applicability in real-world scenarios. Project website: `https://exdex1.github.io/ExDex/`.

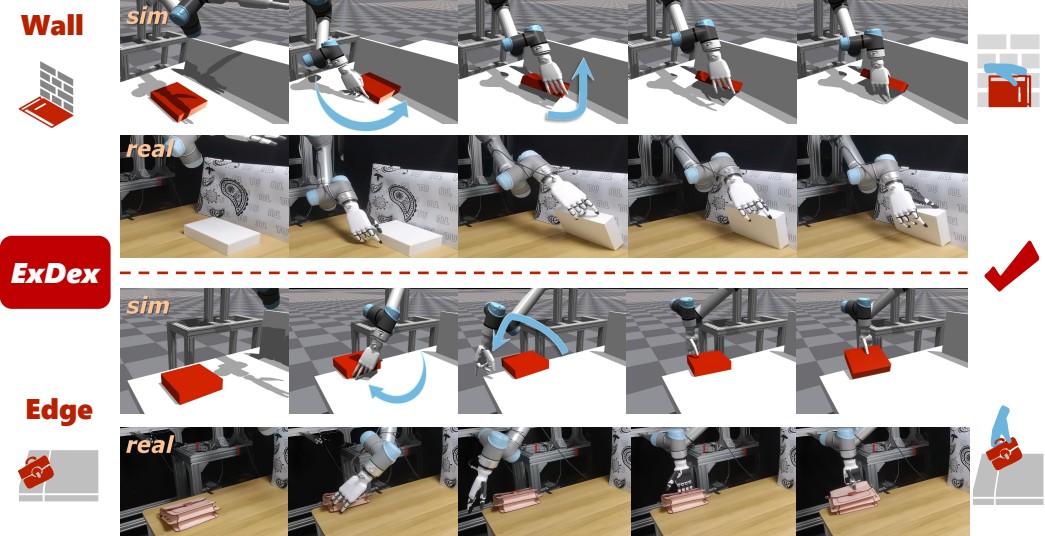

Figure 1: Our framework **ExDex** is demonstrated through two representative tasks: **Wall**, where objects are pushed against the wall so that they can be flipped up and grasped. And **Edge**, where objects are repositioned to table edges, allowing the hand to maneuver into optimal grasping poses.

## 1 Introduction

Humans naturally manipulate objects with their multi-finger dexterous hands through a rich repertoire of strategies. Beyond direct grasping, humans demonstrate remarkable abilities to exploit environmental features for manipulation. For instance, when encountering large, flat objects placed in the middle of a table that are challenging to grasp directly, humans intuitively leverage environmental constraints

like walls or table edges. They seamlessly combine non-prehensile actions such as pushing, sliding, and pivoting with dexterous manipulation to achieve reliable grasps. This adaptive exploitation of environmental affordances enables humans to handle objects that would otherwise be ungraspable through direct manipulation alone. Such environment-aware manipulation strategies significantly expand the range of objects that can be successfully manipulated, demonstrating the sophisticated interplay between dexterous control and environmental interaction.

Replicating human-like extrinsic dexterity in multi-finger robotic hands remains an unexplored yet crucial problem in robotics. Traditional approaches for dexterous manipulation primarily rely on trajectory optimization with simplified contact models (Chen et al., 2024; Mordatch et al., 2012). However, these methods often fail in contact-rich scenarios due to the complexity of modeling dynamic contact interactions and the uncertainty in physical parameters. While imitation learning has demonstrated promising results in direct dexterous manipulation tasks (Chen et al., 2022d; Shaw et al., 2023; Wang et al., 2024), it faces significant limitations when applied to extrinsic dexterity. The collection of high-quality demonstration data through human teleoperation becomes particularly challenging for dynamic contact-rich manipulations, as operators struggle to precisely control multiple fingers while maintaining stable environmental contacts. Recent years have witnessed remarkable progress in applying reinforcement learning to robotic systems (OpenAI et al., 2019; Yang et al., 2024; Pitz et al., 2023; Handa et al., 2022). Reinforcement learning (RL) provides a powerful framework for training robots in simulated environments before transferring learned policies to real-world applications. By leveraging large-scale parallel simulation, RL enables extensive training without relying on explicit contact modeling or expert demonstrations. This approach is particularly effective for contact-rich manipulation tasks, as it autonomously explores dexterous strategies through reward-driven optimization. Furthermore, large-scale RL training facilitates the emergence of stable and natural behaviors, which are essential for robust real-world deployment. These advantages make RL especially well-suited for mastering complex non-prehensile manipulation strategies that require adaptive control and dynamic environmental interactions.

While existing researches often simplify the problem by placing objects near external contacts and utilize end-to-end reinforcement learning policies training (Zhou & Held, 2023; Chen et al., 2023a), they overlook the complexity of strategic object repositioning and environmental interaction. Effectively leveraging extrinsic dexterity to manipulate ungraspable objects demands a hierarchical framework combining non-prehensile manipulation skills with long-term task planning. However, training individual manipulation skills through reinforcement learning alone presents significant challenges. The high-dimensional action space of multi-finger dexterous hands, combined with the contact-rich nature of environmental interactions, makes each subtask difficult to learn. Moreover, the challenge extends to high-level strategic planning. Optimal object relocation requires consideration of multiple factors: the current object position, available external contacts, robot arm configuration, and kinematic constraints. Simply choosing the nearest point that is convenient to utilize the external environment is insufficient, and often leads to suboptimal or failed manipulations. Instead, the system must evaluate potential target locations while considering the robot's reachability, joint limits, and possible collision-free paths. This intricate planning requirement makes high-level strategic planning particularly challenging, as it must simultaneously account for all these constraints to ensure successful manipulation.

To overcome these challenges, we present **ExDex**, a framework for dexterous manipulation of ungraspable objects using extrinsic dexterity with multi-finger hands, focusing particularly on leveraging walls and table edges. We introduce a hierarchical learning approach combining a high-level planner for identifying optimal environmental contacts with a low-level controller for precise non-prehensile manipulation. The high-level planner generates target positions and transition signals, while the low-level controller executes pushing policies to achieve these poses, followed by grasping policies selected by transition signals based on the external environment. The experiments in both simulation and real-world settings validate our framework's effectiveness, and the results demonstrate successful generalization to unseen objects and zero-shot transfer to physical systems.

In summary, our main contribution encompasses:

- First exploration of extrinsic dexterity with multi-finger dexterous manipulation in both simulation and real-world scenarios.

- Novel hierarchical framework combining high-level planning and low-level control for occluded grasp tasks.

- Extensive experimental validation demonstrating system effectiveness across simulated and physical environments.

## 2 RELATED WORKS

### 2.1 DEXTEROUS MANIPULATION

Multi-finger dexterous manipulation remains a significant challenge in robotics. Traditional trajectory optimization methods based on dynamic models (Chen et al., 2024; Mordatch et al., 2012; Bai & Liu, 2014; Kumar et al., 2014) often fall short due to simplified contact assumptions, especially in complex, contact-rich tasks. Recent research has demonstrated remarkable success with imitation learning (Shaw et al., 2023; Wang et al., 2024; Chen et al., 2022c; Radosavovic et al., 2021; Arunachalam et al., 2023; Handa et al., 2020; Sivakumar et al., 2022; Qin et al., 2023; 2022b; Cui et al., 2022; Haldar et al., 2023; Qin et al., 2022a; Arunachalam et al., 2022; Zhong et al., 2025). 3D-ViTac (Huang et al., 2024) achieves precise manipulation using tactile feedback, while DexCap (Wang et al., 2024) enables complex bimanual tasks through in-the-wild data collection using data gloves. However, these approaches face limitations due to the high cost of human demonstration data and collecting data for highly dynamic actions (such as flipping objects from wall edges). Recently, reinforcement learning (RL) has been widely adopted for dexterous hand manipulation tasks, spanning in-hand object reorientation (Chen et al., 2021; 2022a; Yin et al., 2023; Qi et al., 2023b;a; Dasari et al., 2023; OpenAI et al., 2019; Yang et al., 2024; Pitz et al., 2023; Handa et al., 2022; Khandate et al., 2023), bimanual manipulation (Huang et al., 2023; Lin et al., 2024; Chen et al., 2022b), pre-grasping (Zhou & Held, 2023; Ding et al., 2024), hand long-horizon manipulation (Chen et al., 2023b; Huang et al., 2023). We develop policies that adapt to the dynamic motion control of real robots using RL. To our knowledge, this work represents the first exploration of extrinsic dexterity with dexterous hands demonstrated in both simulation and real-world environments.

### 2.2 EXTRINSIC DEXTERITY

External environmental resources such as contacts, gravity, and dynamic motions(Dafle et al., 2014) enable robot hands to grasp and manipulate objects even without suitable contact points. Previous work has demonstrated the utility of environmental interactions, including external contacts for object grasping(Zhou & Held, 2023; Ma et al., 2024; Ding et al., 2024; Chen et al., 2023a) and reorientation (Stepputtis et al., 2018), as well as leveraging gravity (Dong et al., 2023) or dynamic motions (Ha & Song, 2022; Dafle et al., 2014) to improve grasping postures. However, except for Chen et al. (2023a), these works primarily employ grippers or underactuated multi-finger hands. We instead utilize a five-fingered dexterous hand, leveraging its greater degrees of freedom and flexibility for enhanced grasping capabilities and improved policy generalization across multiple objects. Zhou & Held (2023) present the closest approach to ours, learning a closed-loop RL policy with restrictive assumptions, like objects being initially positioned near walls and walls being sufficiently low for grippers to access objects from above. In contrast, our approach accommodates objects anywhere in the workspace. Moreover, most previous work has not explored how to leverage the high degrees of freedom of dexterous hands for extrinsic dexterity. While UniDexFPM (Wu et al., 2024) investigated dexterous hand pre-grasp manipulation in tabletop environments, their results were limited to simulation without physical robot validation. We propose a hierarchical framework with a high-level planner predicting target external contacts and a low-level controller learning a series of non-prehensile manipulation skills for object relocating and retrieval.

## 3 TASK FORMULATION

In this paper, we address the challenge of grasping ungraspable objects that have large, flat base surfaces using a dexterous multi-finger hand. The task objective is to employ a sequence of non-prehensile manipulations to reposition objects near environmental features that can assist in successful grasping. We formulate this task as a finite horizon Markov Decision Process (MDP), defined by the 5-tuple $(\mathcal{S}, \mathcal{A}, R, P, \gamma)$. Here, $\mathcal{S}$ and $\mathcal{A}$ represent the state and action spaces respectively. The stochastic dynamics $P : \mathcal{S} \times \mathcal{A} \times \mathcal{S} \to [0, 1]$ determine the probability of transitioning to state $s'$ given current state $s$ and action $a$. $R : \mathcal{S} \times \mathcal{A} \times \mathcal{S} \to \mathbb{R}$ defines the reward function, and $\gamma \in (0, 1)$ is the discount factor. Our objective is to train a policy $\pi$ that maximizes the expected cumulative reward $\mathbb{E}_\pi[\sum_{t=0}^{T-1} \gamma^t R]$ in an episode with $T$ time steps.

Figure 2: **Illustration of the ExDex system design.** (A) Training: Our system is trained in three stages: In Stage 1, we train a prediction model $\pi_{\text{pre}}$ through supervised learning. Stage 2 focuses on training three low-level policies via reinforcement learning: $\pi_{\text{push}}$, $\pi_{\text{wall}}$, and $\pi_{\text{edge}}$. In Stage 3, we jointly finetune these policies to ensure better transitions between consecutive skills. (B) Inference: Firstly, $\pi_{\text{pre}}$ is used to predict the target position $P_t$ and a signal $s$. $\pi_{\text{push}}$ then moves the object to the predicted $P_t$, followed by the policy ($\pi_{\text{wall}}$ or $\pi_{\text{edge}}$) selected by the signal $s$ to complete the grasp.

## 4 METHOD

In this section, we introduce our system for dexterous non-prehensile manipulation for ungraspable objects. The overview of the system is shown in Figure 2. Our framework consists of three parts: the High-level Planner Design (Section 4.1), Low-level Policy Training (Section 4.2) and Joint Finetuning (Section 4.3). The details of our sim-to-real policy transfer are introduced in Section 4.4.

### 4.1 HIGH-LEVEL PLANNER DESIGN

The first step in extrinsic dexterity is relocating objects to environments that can be leveraged for manipulation, such as walls or table edges. Therefore, planning a desired location where external conditions can be effectively utilized is crucial for successful extrinsic dexterity. To achieve this, we train a prediction model $\pi_{\text{pre}}$ through supervised learning to predict target positions for object relocation. The model takes environmental point cloud data $p$ as input and outputs three-dimensional target position $P_t = (P_x, P_y, P_z)_t$ and a signal $s$. The predicted $P_t$ serves as the target location to guide the low-level policy $\pi_{\text{push}}$ to achieve object relocation. Subsequently, the signal $s$ helps to pick a low-level policy from $\pi_{\text{wall}}$ and $\pi_{\text{edge}}$ automatically to grasp the object after pushing.

### 4.2 LOW-LEVEL POLICY TRAINING

We train three specialized policies using model-free reinforcement learning: (1) A $\pi_{\text{push}}$ policy that pushes objects to the target position $P_t$ based on the predicted target position $P_t$ from the high-level planner; (2) A $\pi_{\text{wall}}$ policy for grasping objects near walls starting from $P_t$; and (3) A $\pi_{\text{edge}}$ policy for

retrieving objects from table edges at $P_t$. The following subsections detail the observation and action space, reward design, and training strategy.

### 4.2.1 OBSERVATION AND ACTION SPACE

The observation space $\mathcal{S} = \{q_t, \{F_t^{f,i}\}_{i=1}^5, p_t^{\text{obj}}, v_t^{\text{obj}}, P_t, c_p\}$ consists of several components: robot state (proprioceptive arm and hand joint positions $q_t \in \mathbb{R}^{18}$, and five fingertip poses $\{F_t^{f,i}\}_{i=1}^5 \in \mathbb{R}^{15}$), object position $p_t^{\text{obj}} \in \mathbb{R}^3$ and velocity $v_t^{\text{obj}} \in \mathbb{R}^6$, target information (predicted position $P_t \in \mathbb{R}^3$ from high-level planner), and contact information (hand-designed contact position $c_p \in \mathbb{R}^3$ that maintains a fixed relative position to the object center).

The action space $a_t = \{a_t^{\text{arm}}, a_t^{\text{hand}}\}$ consists of two components: hand joint positions $a_t^{\text{hand}} \in \mathbb{R}^6$ and relative arm joint positions $a_t^{\text{arm}} \in \mathbb{R}^6$. For the hand, the policy directly outputs absolute joint angles $a_t^{\text{hand}}$ as target positions. For the arm, the policy generates relative position changes $a_t^{\text{arm}}$, which are added to the current joint angles to obtain target positions. The PD controller then converts these target positions into joint torques for both the arm and hand.

### 4.2.2 REWARD DESIGN

To reduce the complexity of reward shaping, we unify our reward function into three components with a staged reward mechanism. The next stage reward is only calculated when specific conditions are met. Here we use $P(\cdot)$ to represent condition probabilities. Specifically:

$$r = r_{\text{motion}} + r_{\text{pregrasp}} \cdot P(a) + r_{\text{grasp}} \cdot P(b) \tag{1}$$

In the following, we describe each reward component in detail. All reward terms share the same goal of minimizing distances between their arguments, thus we denote these proximity-based functions as $T(\cdot, \cdot)$, which output larger values as their arguments become closer. The specific implementation of $P(\cdot)$, $T(\cdot, \cdot)$ and hyperparameter can be found in Appendix G.1.

**Motion reward $r_{\text{motion}}$.** The motion reward guides either object movement to a target pose or fingertip positioning for manipulation. For $\pi_{\text{push}}$ and $\pi_{\text{wall}}$, it encourages the object to reach specific target positions: $r_{\text{motion}} = T(P_t^{obj}, P_t^{target})$. In $\pi_{\text{push}}$, $P_t^{target}$ is set to the position $P_t$ predicted by the high-level planner, while in $\pi_{\text{wall}}$, $P_t^{target}$ is a pre-defined pose above the object to facilitate extrinsic dexterity. For $\pi_{\text{edge}}$, the reward guides fingertip positioning: $r_{\text{motion}} = T(\{F_t^{f,i}\}_{i=1}^5, P_t^{target})$, encouraging the thumb to stay above the object while positioning the other four fingers beneath it.

**Pre-grasp reward $r_{\text{pregrasp}}$.** The pre-grasp reward encourages the hand to achieve an advantageous pre-grasp pose after object repositioning: $r_{\text{pregrasp}} = T(F_t^{f,3}, c_p)$, where $F_t^{f,3}$ is the position of the middle fingertip, and $c_p$ is a relative fixed point on the object.

**Grasp reward $r_{\text{grasp}}$.** Once reaching the pre-grasp position, the grasp reward promotes stable grasping by optimizing fingertip positions relative to the object: $r_{\text{grasp}} = T(P_t^m, P_t^{obj})$, where $P_t^m = \frac{F_t^{f,1} + F_t^{f,3}}{2}$ represents the midpoint of the thumb ($F_t^{f,1}$) and middle fingertip ($F_t^{f,3}$) positions.

### 4.2.3 TRAINING STRATEGY

We employ PPO (Schulman et al., 2017) to train low-level policies, leveraging its stability and sample efficiency. The training process is accelerated through parallel simulations in IsaacGym, enabling simultaneous training across 4096 environments. To improve policy robustness, we incorporate comprehensive domain randomization techniques, including variations in robot and object properties (Appendix G.3). Furthermore, we adopt a curriculum learning strategy to enhance training efficiency. Training begins with some similar objects (Figure 3 (a-pretrain)) at a fixed initial pose. As the success rate improves, we gradually increase task complexity by introducing objects with a larger difference in size (Figure 3 (a-finetune)) and randomizing their initial poses. This progressive learning approach helps policies develop robust manipulation skills while maintaining stable training dynamics.

### 4.3 JOINT FINETUNING

Sequentially executing trained policies often leads to poor performance in long-horizon manipulation tasks. This is primarily because the terminal state of the previous policy may not align well with the initial state distribution of the subsequent policy. This challenge, known as the skill chaining

problem (Chen et al., 2023b; Konidaris & Barto, 2009; Sutton et al., 1999), requires special consideration in policy training. To address this issue, we jointly finetune our policies with the following order. Firstly, to enhance the robustness of $\pi_{\text{push}}$ against potential biases in the high-level planner's predictions, we introduce Gaussian noise with a specified standard deviation to the target position $P_t$ in the observation. Then, to better align the transition states between sequential non-prehensile skills, we replace the initial states of both $\pi_{\text{wall}}$ and $\pi_{\text{edge}}$ with terminal states obtained from $\pi_{\text{push}}$ rollouts. Finally, to improve the capability of the high level planner $\pi_{\text{pre}}$ to predict the target position with a higher likelihood of success, and reduce the prediction of $P_t$ that the robotic arm cannot reach, we refine it with the predicted $P_t$ recorded from successful rollouts for each task. This approach ensures smoother transitions between different manipulation phases, enhancing the overall task performance.

## 4.4 SIM-TO-REAL TRANSFER

When applying the RL policy to the real-world, some environment states can not be estimated accurately like object velocity and fingertip positions. Therefore, we follow the teacher-student distilling framework (Chen et al., 2022a; 2021) to zero-shot transfer our simulation policy into the real dexterous arm-hand system. Specifically, we rollout our policies in simulation sequentially to collect the whole teacher demonstration trajectories. For distillation from demonstration, we employ a transformer-based imitation learning network to predict the target arm-hand joint angles. To mitigate the observation gap between simulation and real-world, the distilled student policy only takes the low dimension state observation including proprioception and object 6d pose as input. To obtain object 6d pose, we use the Segment Anything model (Kirillov et al., 2023) to get the initial mask of the object, followed by FoundationPose (Wen et al., 2023) for pose estimation and tracking. We also build a digital twin framework for sim-to-real transfer. More details about the teacher-student distillation and digital twin can be found in Appendix H.

## 5 EXPERIMENT

In this section, we comprehensively evaluate the performance of our proposed framework in simulation and real-world settings to address these questions: (1) Can our high-level planner generate an optimal object relocating strategy given different external environments? (Section 5.2) (2) Is the dexterous hand motion learned by our low-level polices necessary for our tasks? (Section 5.3) (3) Is our reward design suitable for the non-prehensile manipulation skill training? (Section 5.3) (4) Can our joint finetuning strategy improve the generalizability and robustness of our framework? (Section 5.3) (5) Can our framework learned in simulation be applied to a real-world system? (Section 5.4)

First, we introduce the main setup of our experiments including the dataset, evaluation metric and several baselines for comparison with our method. Then we evaluate the effectiveness of our framework separately from high-level and low-level parts through quantitative and qualitative results. Finally, we provide details of how we conduct real-world experiments and the performance of our method. Our simulation and real-world settings are shown in Figure 3. All simulation results in tables are evaluated in 3 different seeds, and the real-world results are evaluated in 10 trials for each object. Besides, we present more results of environment generalizability and failure case in Appendix B, E.

### 5.1 SETUP

**Dataset.** In simulation environments, we only use boxes with various physics properties as the training asset. We evaluate the generalizability of our framework on 21 other objects with diverse geometries. In the real-world scenario, we conduct an evaluation on 10 objects with different sizes, shapes and physical properties. The objects used in simulation and real-world are shown in Figure 3.

**Evaluation and Metric.** We evaluate the performance of our framework in a scene (Figure 3(c)), containing three tasks utilizing external contacts: Wall, FrontEdge and LeftEdge. FrontEdge and LeftEdge are variants of Edge task but use different directions.Objects are initially placed randomly in the center of the table. We train individual policies for each task as follows. For the Wall task, the policy is trained to first push the object to a wall-adjacent position before utilizing the wall to assist in grasping. For the FrontEdge and LeftEdge task, we follow the training paradigm of Edge task, where the object is pushed to the respective table edge prior to maneuvering the hand to an appropriate position for grasping. We introduce the following metrics for evaluation: (1) *Target Transition Error* (TTE) is the Euclidean distance (cm) between predicted and ground truth target positions. (2) *Success Rate* (SR) is the percentage of successful grasping after a series of non-prehensile manipulation. We define success as the object being grasped steadily above a height threshold (10 cm).

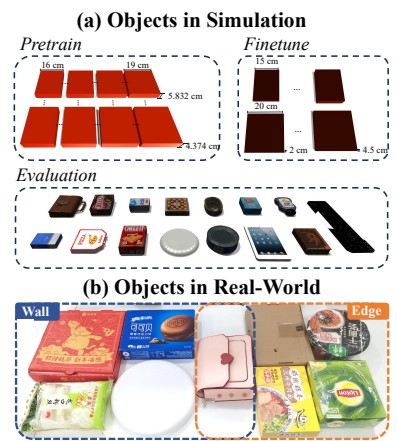

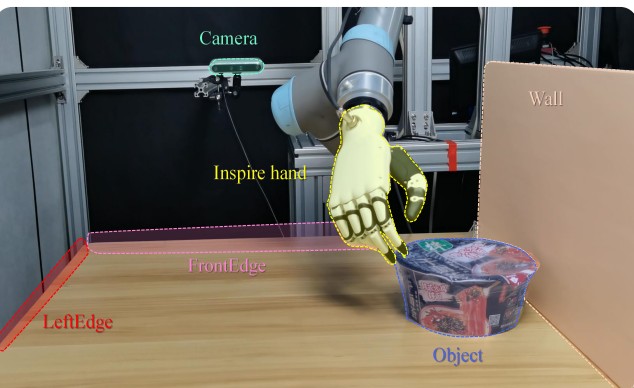

Figure 3: **Overview of the environment setups.** (a) Object sets used in simulation. Policies are first trained on the pretrain set, then finetuned on the finetune set, and finally tested for zero-shot generalization on the evaluation set. (b) Real-world test objects. (c) Workspace of the real-world. We use an Inspired Hand mounted on a UR5e robot, equipped with a RealSense D455 camera.

**Baseline.** We compare our methods with the following baselines and ablations. (1) *Random Target*. Our low-level policies guided by random $P_t$ selection instead of our high-level planner. (2) *Arm-Only*. Our low-level policies trained with arm control only. The hand joint angle is fixed. (3) *Heuristic*. Using predefined action primitives as low-level policies. (4) *Ours w/o MR*. Our low-level policies trained without motion reward. (5) *Ours w/o ST*. Our low-level policies trained without stage reward mechanism. (6) *Ours w/o JH*. Our low-level policies without joint finetuning for the high-level planner. (7) *Ours w/o JL*. Our low-level policies without joint finetuning for the low-level policies. We present more baseline results including learning-based method (Zhou & Held, 2023) and model-based planning MPPI (Pezzato et al., 2025) in Appendix C.

## 5.2 HIGH-LEVEL PLANNER

Table 1: Quantitative comparison of the high-level planner.

| Task | | Random Target | | Ours | |
|------|------|------|------|------|------|
| | | TTE | SR | TTE | SR |
| Wall | Seen | $45.19_{\pm1.94}$ | $66.21_{\pm0.96}$ | $\mathbf{0.41}_{\pm0.00}$ | $\mathbf{83.25}_{\pm0.34}$ |
| | Unseen | $45.15_{\pm1.90}$ | $54.50_{\pm1.47}$ | $\mathbf{0.41}_{\pm0.01}$ | $\mathbf{54.94}_{\pm1.57}$ |
| FrontEdge | Seen | $31.63_{\pm2.77}$ | $88.23_{\pm1.04}$ | $\mathbf{3.16}_{\pm0.03}$ | $\mathbf{89.43}_{\pm0.68}$ |
| | Unseen | $31.75_{\pm2.87}$ | $60.11_{\pm2.21}$ | $\mathbf{2.85}_{\pm0.06}$ | $\mathbf{68.00}_{\pm3.55}$ |
| LeftEdge | Seen | $31.23_{\pm1.23}$ | $74.28_{\pm0.12}$ | $\mathbf{2.58}_{\pm0.01}$ | $\mathbf{76.75}_{\pm2.41}$ |
| | Unseen | $31.32_{\pm1.24}$ | $\mathbf{57.22}_{\pm2.70}$ | $\mathbf{2.97}_{\pm0.07}$ | $54.39_{\pm2.27}$ |

To evaluate the generalizability of the high-level planner for different external contacts, we compare the relocating strategy of our high-level planner with *Random Target*, which randomly produces $P_t$ by the wall or table edge. The quantitative results in Table 1 shows that our high-level planner generates superior relocating positions with lower target transition error (TTE) across all tasks and objects, which facilitates the continual non-prehensile manipulation skills with a higher success rate (SR).

## 5.3 LOW-LEVEL CONTROLLER

**Dexterous hand motion.** We compare our method with *Arm-Only* and *Heuristic* to validate the importance of dexterous hand motion for non-prehensile manipulation. As evidenced in Table 2, our approach demonstrates consistent superiority over both baseline methods across all task configurations. The most notable

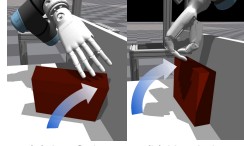
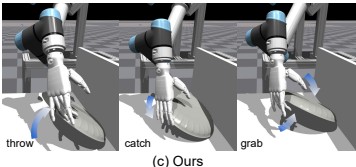

Figure 4: Comparison of *Arm-Only*, *Heuristic*, and *Ours* for Wall task. (a) *Arm-Only*. (b) *Heuristic*. (c) *Ours*.

performance gap emerges in the Wall task, where *Arm-Only* and *Heuristic* exhibit fundamental limitations due to their constrained manipulation strategies. *Arm-Only* can only pivot the object upright like Figure 4(a), causing failure in unseen non box-shaped objects. *Heuristic* follows a circular arc trajectory centered on the midpoint of the contact line between the corner and the object. However, it can only rotate the object to squeeze up against the wall, which prevents stable grasping (Figure 4(b)). Our method overcomes these limitations through the learned dexterous hand motion. Specifically, our RL policy $\pi_{\text{wall}}$ leverages finger motions to lift and dynamically catch the object mid-air (Figure 4(c)), demonstrating superior dexterity. This advantage extends to FrontEdge and LeftEdge task, where our approach maintains robust performance across both seen and unseen objects.

Table 2: Quantitative success rate (SR) comparison of the low-level controller.

| Method | Wall | | FrontEdge | | LeftEdge | |
|---|---|---|---|---|---|---|
| | Seen | Unseen | Seen | Unseen | Seen | Unseen |
| ArmOnly. | $16.27_{\pm0.18}$ | $2.50_{\pm1.22}$ | $29.95_{\pm0.25}$ | $26.39_{\pm2.74}$ | $14.28_{\pm2.39}$ | $19.61_{\pm0.70}$ |
| Heuristic. | $8.03_{\pm0.30}$ | $2.72_{\pm0.61}$ | $73.70_{\pm0.61}$ | $56.61_{\pm1.86}$ | $63.91_{\pm1.22}$ | $48.61_{\pm0.98}$ |
| w/o MR. | $0.00_{\pm0.00}$ | $0.17_{\pm0.14}$ | $0.00_{\pm0.00}$ | $1.00_{\pm0.14}$ | $0.00_{\pm0.00}$ | $0.01_{\pm0.02}$ |
| w/o ST. | $0.01_{\pm0.02}$ | $0.22_{\pm0.21}$ | $2.28_{\pm0.24}$ | $5.44_{\pm0.42}$ | $0.56_{\pm0.31}$ | $1.36_{\pm0.14}$ |
| w/o JL. | $49.47_{\pm3.67}$ | $19.39_{\pm2.91}$ | $74.56_{\pm3.31}$ | $32.61_{\pm2.21}$ | $48.18_{\pm0.47}$ | $12.94_{\pm0.55}$ |
| w/o JH. | $\mathbf{83.88}_{\pm0.33}$ | $54.28_{\pm0.97}$ | $82.02_{\pm0.92}$ | $63.22_{\pm3.80}$ | $73.33_{\pm2.70}$ | $53.44_{\pm3.91}$ |
| **Ours** | $83.25_{\pm0.34}$ | $\mathbf{54.94}_{\pm1.57}$ | $\mathbf{89.43}_{\pm0.68}$ | $\mathbf{68.00}_{\pm3.55}$ | $\mathbf{76.75}_{\pm2.41}$ | $\mathbf{54.39}_{\pm2.27}$ |

**Reward Design.** To investigate the importance of our reward design in low-level policy learning, we conduct ablation studies on two key components: (1) the motion reward (*Ours w/o MR*), which guide the object toward the target pose, and (2) the stage reward mechanism (*Ours w/o ST*), which dynamically adjust different reward components during training. As shown in Table 2, removing the motion reward (*Ours w/o MR*) leads to near zero success rate across all tasks, demonstrating that precise motion reward guidance is essential for low-level policy training. Similarly, ablating the stage reward mechanism (*Ours w/o ST*) causes a drastic performance drop, which confirms that dynamic reward adjustment is critical for smooth transitions between task stages.

**Joint Finetuning.** To assess the effectiveness of our joint finetuning approach in enhancing the framework's generalization capability and robustness, we conducted systematic ablation studies examining both the high-level planner (*Ours w/o JH*) and low-level controller (*Ours w/o JL*) components. The experimental results in Table 2 reveal that removing joint finetuning for the low-level controller results in substantial performance degradation, particularly on unseen objects. Specifically, SR drops approaching 40% in these cases, clearly demonstrating that our joint finetuning approach effectively bridges the gap of the state mismatch of chaining low-level policies. While the baseline configuration without high-level planner finetuning (*Ours w/o JH*) maintains reasonable performance across all tasks, our analysis shows that incorporating target positions from successful demonstrations to refine the high-level planner yields consistent slight performance improvements. This suggests that both components of our joint finetuning strategy contribute to the framework's overall effectiveness.

## 5.4 REAL-WORLD EXPERIMENT

Table 3: Results for Real-world Experiments using teacher-student distillation

| Size ($cm^3$) | 16.5x15.1x6.2 | 17.3x17.3x7.5 | 23x16.2x5 | 20.7x16.5x7 | 19x14x4 |
|---|---|---|---|---|---|
| Wall | 10/10 | 10/10 | 9/10 | 8/10 | 8/10 |
| Size ($cm^3$) | 21x13x4.4 | 22x16x4.4 | 24.7x23.8x4 | 20.7x16.5x7 | 23.5x23.5x2.3 |
| FrontEdge | 7/10 | 9/10 | 9/10 | 6/10 | 8/10 |
| Size ($cm^3$) | 21x13x4.4 | 22x16x4.4 | 24.7x23.8x4 | 20.7x16.5x7 | 23.5x23.5x2.3 |
| LeftEdge | 5/10 | 7/10 | 8/10 | 5/10 | 7/10 |

**Hardware Setup.** We set up the identical scenario in the real-world as in the simulation, one multi-finger Inspire Hand mounted on a UR5e robot for our experiments, as shown in Figure 3. To obtain the real-time visual observation for object pose estimation and environment state extraction, we use a single RealSense D455 depth camera.

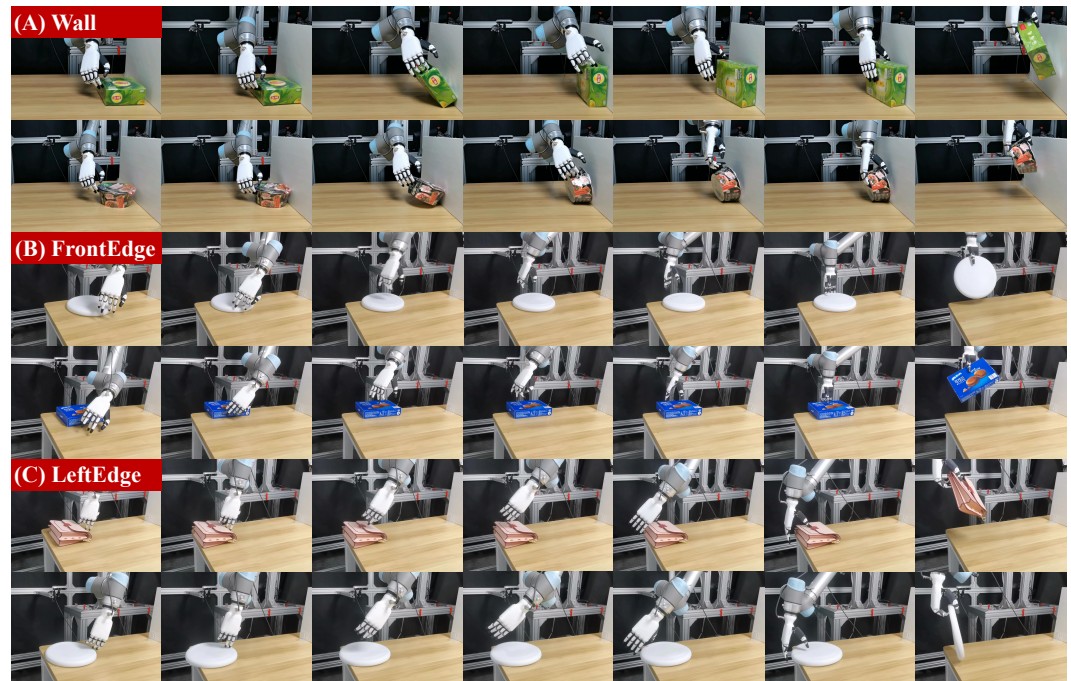

Figure 5: Real-world experiment demonstrations. The snapshots show successful executions of our framework on various objects. (a) Wall tasks. (b) FrontEdge tasks. (c) LeftEdge tasks.

**Evaluation and Metric.** We evaluate our framework in real-world scenarios following the same protocols as our simulation experiments. For all tasks, objects are initially placed randomly in the center of the table. We use the success rate (SR) as our evaluation metric, following the same criteria defined in Section 5.1, where a grasp is considered successful if the object is lifted steadily above a specified height threshold (20cm).

**Object set.** To evaluate the sim-to-real transfer capability of our framework, we conduct experiments on a diverse set of real-world objects that vary significantly in their physical properties. Our test objects, illustrated in Figure 3, include items with different geometries, sizes, and masses. Moreover, we deliberately include several deformable objects, which present additional challenges for non-prehensile manipulation due to their changing dynamics during interaction.

**Sim-to-real performance.** As shown in Table 11, our policies achieve robust performance on real-world objects, with success rates exceeding 80% for most tasks, demonstrating effective sim-to-real transfer capability. More importantly, our framework maintains high success rates even when handling objects that differ significantly from the simulation training set in terms of size and physical properties. This robust performance extends to challenging scenarios involving deformable objects, whose dynamics are particularly difficult to model accurately in simulation. The detailed visualization of our real-world experiments is shown in Figure 5.

## 6 CONCLUSION

In this work, we investigate the challenging problem of manipulating ungraspable objects using extrinsic dexterity with a multi-finger hand. We present a hierarchical framework that combines strategic planning with dexterous manipulation skills. Our framework features a high-level planner that intelligently selects optimal external contacts and predicts target positions, coupled with a low-level controller that executes precise non-prehensile manipulation skills. Through extensive experiments in simulation, we demonstrate our framework's superior performance across different external contacts and various objects. Moreover, the successful transfer of our policies from simulation to a real-world robot system validates the practical applicability of our method.

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

## A LIMITATION

There are several limitations of our work. (1) *Limited operation space*. Our current implementation relies on a fixed-base robot, which constrains the workspace to a corner region of the table. A potential solution would be to integrate our framework with a mobile manipulator. (2) *Clutter scene generalizability*. All of our experiments are conducted on a clean table which only contains wall or target objects. Future work could focus on enhancing both our prediction model and pushing policy to enable robust object repositioning in cluttered environments.

## B ENVIRONMENT CONFIGURATION GENERALIZABILITY

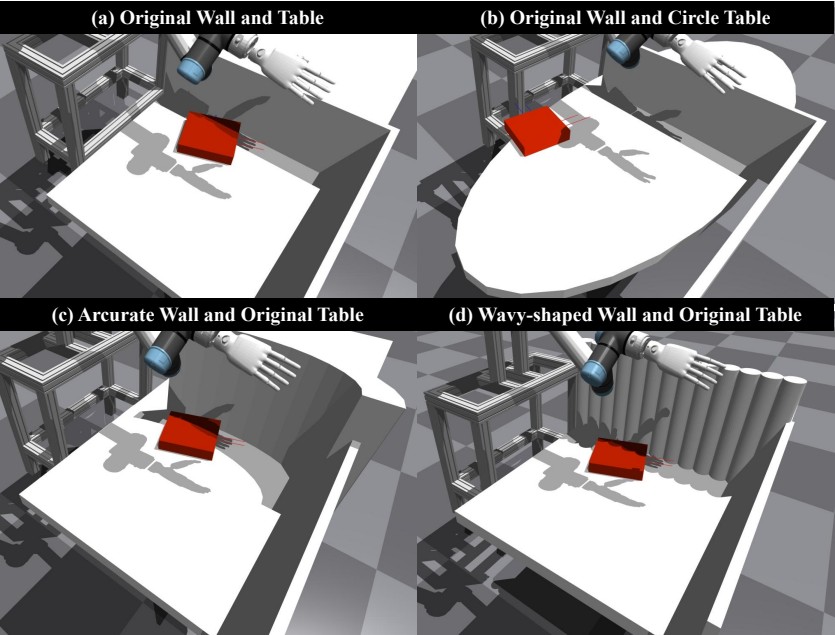

Figure 6: **Different environment configuration**

To assess the generalizability of our framework across diverse environmental configurations, we conduct experiments involving various contact geometries, including arcuate walls, wavy-shaped walls, and circular tables as illusrtrated in Figure 6. The experimental results, presented in the Table 4, demonstrate our framework's robust performance across these geometrically varied contact scenarios, confirming its adaptability to different environmental configurations.

Table 4: Evaluation of generalizability for different environment configurations.

| Environment configuration | Seen | Unseen |
|---|---|---|
| arcurate wall | 81.28 | 51.00 |
| wavy-shaped wall | 70.76 | 47.17 |
| circle table | 93.55 | 63.83 |

## C MORE BASELINE RESULTS

**Zhou et al.'s method.** Zhou & Held (2023) trains an RL policy for the Wall task using a parallel gripper. Its reward function includes two items to guide the parallel gripper to pivot the object up leveraging the wall contact. The first item encourages the end-effector to align with the target grasp pose predefined in the object frame, while the second item encourages the object to be rotated up by penalizing when the six manually defined points on the end-effector locate lower than the table surface. We adapts the reward function into our experiment settings with several modifications. Specifically, the target grasp pose in the first item is redefined by five fingers target positions and

palm rotation. Besides, for the penalty item, we replace the six predefined points with five fingertip positions, penalizing when the five fingertips of the target grasping pose locate below the table.

Table 5: Experimental results of Learning-based method

| Method | Wall | |
|---|---|---|
| | Seen | Unseen |
| Zhou & Held (2023) | 42.30 | 14.17 |

This method achieves success rates of **42.30%** and **14.17%** on seen and unseen objects respectively, both lower than our method, as shown in Table 5. We observed that our policy picks up the object to a certain height and then inserts the finger under the object, while the policy trained by Zhou & Held (2023) tends to find a lower energy strategy, that is, to gently pick the object just enough to insert the fingers under it, so that the object takes less time to take off, resulting in a greater probability of failure to insert after picking up.

**MPPI.** We employ a model-based planning controller named Model Predictive Path Integral (MPPI) with IsaacGym serving as the underlying dynamics model. Our implementation leverages a dual-threaded architecture, consisting of two parallel IsaacGym simulations: the Planner and the World.

- Planner Thread: This thread generates noise sequences by exploiting IsaacGym's large-scale parallel simulation capabilities. These sequences are weighted via importance sampling, where the weights are derived from a state-dependent cost function. We define the cost function as follows, which is the negation of our reward function.

$$r = -(r_{\text{motion}} + r_{\text{pregrasp}} \cdot P(a) + r_{\text{grasp}} \cdot P(b)) \qquad (2)$$

- World Thread: This thread executes the approximately optimal control sequence, computed as the weighted average of the sampled trajectories. For each trajectory, the method weights it higher when it has a lower cost (a higher reward). To ensure real-time asynchronous coordination, the Planner periodically updates its state from the World, while the World executes the action sequences computed by the Planner.

Table 6: Experimental results of model-based planning

| Method | Wall | Front Edge |
|---|---|---|
| MPPI (Pezzato et al., 2025) | 22.38 | 7.62 |

We integrate the MPPI method into the relocation stage $\pi_{push}$ of our framework for both Wall and Edge task. Due to the difficulty in inserting the finger under the object in the Wall task which leads to the failure on all objects, we simplify the metric and consider it a success as long as the object is rotated up to the center point by 2cm. For the Edge task, we extend the time limit of the grasping stage from 450 steps to 1000 steps. We evaluate its performance only on the test set for fair comparison, for our policy is trained on the training set while the MPPI is not. The results are shown in Table 6

## D    MORE OBJECTS OF TEST SET

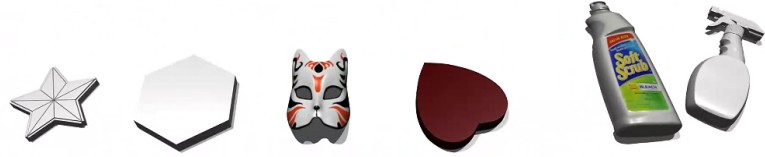

Figure 7: **More Objects of test set with greater geometric difference.**

# E    FAILURE CASE

To reveal the application of our system for diverse objects, we manually categorize these objects into four classes (Flat, Box, Cylinder + Bottle and Irregular) based on the size and geometry and do failure case analysis on each class for the Wall and FrontEdge task (Table 7). We list 6 failure cases for Wall task and 5 failure cases for FrontEdge task respectively. Besides, we test 10 times for each object and average the failure times for each case.

For the Wall task, cylindrical and bottle-like objects demonstrate particularly challenging dynamics, as evidenced by their timeout rates (3.0 instances) and pose deviation failures (1.6 instances). Box-shaped objects exhibit notable difficulties with upright posture maintenance (2.5 instances), while showing relative robustness in other failure categories. Flat objects present moderate performance across most metrics, with timeout being the most prevalent issue (1.4 instances). Irregular objects display a unique profile with significant pickup failure rates (1.4 instances), suggesting specific handling challenges for this category.

For the FrontEdge task, Flat objects, due to their large planar areas with very small thickness, show pronounced vulnerability to contact knock-off events (3.6 instances) and edge push-off failures (1.4 instances). Cylindrical objects fail most in contact knock-off and timeout cases (1.8 instances each). Box-shaped objects demonstrate remarkable robustness in this configuration, with minimal failures across all measured parameters. Irregular objects continue to present handling difficulties, particularly with timeout (2.2 instances) and grasp slippage (1.2 instances) failures.

Table 7: Failure case on test set in 4 classes for Wall and FrontEdge task

| Wall | Pose Deviation | Pickup | Timeout | Upright Posture | Edge Slippage | Collision Ejection |
|---|---|---|---|---|---|---|
| Flat | 0.8 | 0.2 | 1.4 | 0.2 | 0.4 | 0 |
| Box | 0.17 | 0 | 1.83 | 2.5 | 0.17 | 0.5 |
| Cylindrical | 1.6 | 0.4 | 3 | 0.2 | 0 | 0.2 |
| Irregular | 1 | 1.4 | 1 | 0.2 | 0.2 | 0 |

| FrontEdge | Contact Knock-off | Edge Push-off | Timeout | Grasp Slippage | Push Failure |
|---|---|---|---|---|---|
| Flat | 3.6 | 1.4 | 0.8 | 0 | 1.2 |
| Box | 0.17 | 0 | 0.17 | 0.33 | 0 |
| Cylindrical | 1.8 | 0.4 | 1.8 | 0.8 | 0 |
| Irregular | 0.6 | 0.8 | 2.2 | 1.2 | 0.4 |

# F    OVERALL SUCCESS RATE

Given various environment constraints, determining which constraint an object can utilize to reach a higher success rate is precisely one of the goals we intend to achieve by designing the high-level planner $\pi_{pre}$, which outputs a signal $s$ to choose a low-level policy from $\pi_{wall}$ and $\pi_{edge}$ as we mentioned in Sec 4.1. During the joint finetuning (Sec 4.3) process, we collect 3000 data respectively for the Wall and Edge tasks and use the successful ones to finetune $\pi_{pre}$. Specifically, if the success rate of manipulating a specific object at the table edge is higher, more data at the table edge would be collected, leading $\pi_{pre}$ to be more inclined to output the table edge signal when encountering that object. Since the data for training and finetuning $\pi_{pre}$ only includes our original wall and edge, we test the overall success rates of each object when those two constraints are present, relying on $\pi_{pre}$ to select which environment to utilize, and sort them into the four categories.

Table 8: The overall success rates.

| Category | Success rate |
|---|---|
| Flat | 55.75 |
| Box | 74.46 |
| Cylindrical | 54.25 |
| Irregular | 40.94 |

Table 8 shows that the overall success rate is almost consistent with the average success rate of individual Wall task and Edge task, rather than the maximum value among them. We find that when randomly placing an object on the table, $\pi_{pre}$ predicts the signal $s$ with equal probability for the Wall and Edge task. Specifically, when predicting $s$, $\pi_{pre}$ is more inclined to refer to the distance between the object and the environment, and also the configuration of the robot arm. For instance, due to the configuration limitations of the robot arm, when pushing the object which is close to the table edge to the wall, certain points in the movement path of the end effector are unattainable for the end of the ur5e robot arm we use.

## G  TRAINING DETAILS

### G.1  REWARD DESIGN

In Equation 2 , we divide the reward function into three parts: $r_{\text{motion}}$, $r_{\text{pregrasp}}$, and $r_{\text{grasp}}$. $r_{\text{motion}}$ guides the policy toward its ultimate goal and remains active throughout the entire task execution. $r_{\text{pregrasp}}$ encourages the dexterous hand to move towards the object for pre-grasp following the trajectory we expect (first moving towards $c_p$, and then moving towards the object center $P_t^{obj}$). $r_{\text{grasp}}$ is designed to facilitate successful object grasping after pre-grasp. These three parts of the reward function are illustrated in Figure 8(a).

In this section, we explain items that are related to the reward function in detail, then describe our reward design for each task respectively.

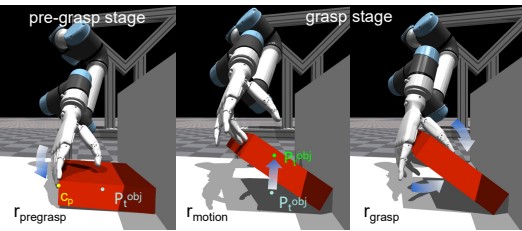

(a) Three parts of the reward function

(b) $c_p$ position for each task (c) Importance of $c_p$

Figure 8: **Illustration for reward design.** (a) Three parts of the reward function. (b) $c_p$ position for each task. (c) Importance of $c_p$.

#### G.1.1  REWARD ITEMS

**Contact point $c_p$.**  The contact point $c_p \in \mathbb{R}^3$ is defined as a fixed spatial offset from the center point of all objects, maintaining a constant distance along the object's width axis as show in Figure 8(a) without manually designed for each object. To be specific, given an object with its center point $P_t^{obj} = (x, y, z)$, the contact point is computed as $c_p = P_t^{obj} + d \cdot \hat{w}$, where $d = 7cm$ represents the fixed offset distance and $\hat{w}$ denotes the unit vector along the object's width dimension. In the pre-grasp stage, we hope that the hand approaches the object from its side for both $\pi_{\text{wall}}$ and $\pi_{\text{push}}$, instead of above where actually obtains a higher $r_{\text{motion}}$. The specific position of $c_p$ and its importance are shown in Figure 8 (b, c).

**Distance function $T(\cdot, \cdot)$.**  The distance function $T(\cdot, \cdot)$ is formulated as $T(a, b) = e^{-\|a-b\|_2}$. $T(a, b)$ is used to measure the distance between its arguments, and increases as the distance difference between $a$ and $b$ decreases.

#### G.1.2  WALL

For $\pi_{\text{wall}}$, we expect that the hand first approaches the object from its side guided by $r_{\text{pregrasp}}$, and then grasp it between the thumb and other fingers guided by $r_{\text{grasp}}$. Therefore, we set $P(a) = 1, P(b) = 0$ in the pre-grasp stage and switch to $P(a) = 0, P(b) = 1$ in the grasp stage. The switch occurs when $\|F_t^{f,3} - c_p\|_2 < 3cm$ which indicates sufficient proximity between the middle finger $F_t^{f,3}$ and contact point $c_p$. Besides, $r_{\text{motion}}$ continuously compute the distance between the object $P_t^{obj}$ and a point $P_t^{target} = P_t^{obj} + [0, 0, 10]^T (cm)$. Notice that this point is located above the object, which is designed to guide the robot to rotate up the object. The specific representation of the reward function for $\pi_{\text{wall}}$ is:

$$\begin{cases} r = T(P_t^{obj}, P_t^{target}) + T(F_t^{f,3}, c_p), & \|F_t^{f,3} - c_p\|_2 \geq 3cm \\ r = T(P_t^{obj}, P_t^{target}) + T(P_t^m, P_t^{obj}), & \|F_t^{f,3} - c_p\|_2 < 3cm \end{cases} \quad (3)$$

### G.1.3 EDGE

For $\pi_{\text{edge}}$, the object is already pushed to expose a graspable side at the table edge, so there is no need for the finger to get to $c_p$, leading $P(a) \equiv 0$. Also, the object $P_t^{obj}$ no longer needs to be moved to any other position, allowing the policy to focus exclusively on vertical finger coordination, keeping the thumb above the object while the other fingers below the object. The motion reward is formulated as: $r_{\text{motion}} = T(\{F_t^{f,i}\}_{i=1}^5, P_t^{target}) = e^{-\sum_{i=1}^5 \|F_t^{f,i} - P_t^{target}\|_2}$ , where target heights are:

$$P_t^{target} = \begin{cases} P_t^{obj} + [0, 0, 0.15](\text{cm}), & \text{for thumb finger} \\ P_t^{obj} - [0, 0, 0.05](\text{cm}), & \text{for other fingers} \end{cases} \tag{4}$$

We set $P(b) = 1$ if the middle finger moves below the object to activate the grasp stage, encouraging the midpoint of the thumb and middle finger $P_t^m$ to get close to the object center point $P_t^{obj}$. Otherwise, we maintain $P(b) = 0$.

$$\begin{cases} r = T(\{F_t^{f,i}\}_{i=1}^5, P_t^{target}), & F_{t,z}^{f,3} \geq P_{t,z}^{obj} \\ r = T(\{F_t^{f,i}\}_{i=1}^5, P_t^{target}) + T(P_t^m, P_t^{obj}), & F_{t,z}^{f,3} < P_{t,z}^{obj} \end{cases} \tag{5}$$

### G.1.4 PUSH

For $\pi_{\text{push}}$, staged reward and grasping are unnecessary, resulting in $P(a) \equiv 1$ and $P(b) \equiv 0$. $r_{\text{pregrasp}}$ guides the hand to push the object from its side, and $r_{\text{motion}}$ narrows the gap between the object $P_t^{obj}$ and target position $P_t$ predicted by our high-level planner.

$$r = T(P_t^{obj}, P_t) + T(F_t^{f,3}, c_p) \tag{6}$$

### G.2 OBJECT ASSETS

For constructing the object assets depicted in Figure 3(a), we procedurally generate randomized boxes for pretraining and finetuning by applying domain randomization to their sizes and masses as shown in Table 9. To diversify our evaluation objects, we carefully selected large, flat-shaped models from existing datasets (Xiang et al., 2020; Calli et al., 2015) and 3D model websites (Google; Google LLC; CGTrader, Inc.).

### G.3 DOMAIN RANDOMIZATION

We apply domain randomization in the simulation environment to improve the robustness of our policy. The detailed parameters are illustrated in Table 9.

### G.4 HYPERPARAMETERS OF THE PPO

Table 10 shows the hyperparameters of the PPO.

### G.5 COMPUTE RESOURCES DETAILS

Our implementation utilizes PyTorch as the deep learning framework. All experiments were conducted on a Ubuntu 20.04 system equipped with a single NVIDIA GeForce RTX 4090 GPU (24GB memory), used for both training and inference phases. Each policy was trained for 100,000 epochs on average, requiring approximately 8 hours of computation time per policy.

## H SIM-TO-REAL DETAILS

### H.1 TEACHER-STUDENT DISTILLATION.

We collect 1000 demonstration trajectories with the teacher RL policies in simulation for each task. Here we manually design some rules to remove the unnatural or dangerous behaviors that emerge from exploiting the simulator dynamics, but don't transfer well to real-world. We use a transformer-based network to imitate the curated demonstration. The network architecture is as followed.

The network takes as input a sequence of concatenated state observations (dimension: 13) spanning 10 historical frames. An initial feature extraction module processes each frame independently through two linear layers (128 and 512 units respectively), each followed by ReLU activation and layer normalization. The extracted features are augmented with learnable positional encodings to preserve

Table 9: Domain randomization of Wall, FrontEdge and LeftEgde task. The units of measurement are as follows: length in meters (m), mass in kilograms (kg), and angles in radians (rad).

| Parameter | Type | Distribution | Initial Range |
|---|---|---|---|
| **Robot** | | | |
| Mass | Scaling | uniform | [0.5, 1.5] |
| Friction | Scaling | uniform | [0.7, 1.3] |
| Joint Lower Limit | Scaling | loguniform | [0.0, 0.01] |
| Joint Upper Limit | Scaling | loguniform | [0.0, 0.01] |
| Joint Stiffness | Scaling | loguniform | [0.0, 0.01] |
| Joint Damping | Scaling | loguniform | [0.0, 0.01] |
| **Object** | | | |
| Mass | Scaling | uniform | [0.34, 1.26] |
| Friction | Scaling | uniform | [0.5, 1.0] |
| SizeX | Scaling | uniform | [0.15, 0.20] |
| SizeZ | Scaling | uniform | [0.02, 0.06] |
| PositionX Scale | Scaling | uniform | [-0.10, 0.10] |
| PositionY Scale | Scaling | uniform | [-0.15, 0.15] |
| Rotation Scale | Scaling | uniform | [-0.5, 0.5] |
| Position Noise | Additive | gaussian | [0.0, 0.02] |
| Rotation Noise | Additive | gaussian | [0.0, 0.2] |
| **Observation** | | | |
| Obs Correlated. Noise | Additive | gaussian | [0.0, 0.001] |
| Obs Uncorrelated. Noise | Additive | gaussian | [0.0, 0.002] |
| **Action** | | | |
| Action Correlated Noise | Additive | gaussian | [0.0, 0.015] |
| Action Uncorrelated Noise | Additive | gaussian | [0.0, 0.05] |
| **Environment** | | | |
| Gravity | Additive | normal | [0, 0.4] |

Table 10: Hyperparameters of PPO.

| Hyperparameters | Value |
|---|---|
| Num mini-batches | 4 |
| Num opt-epochs | 5 |
| Num episode-length | 8 |
| Hidden size | [1024, 512, 256] |
| Clip range | 0.2 |
| Max grad norm | 1 |
| Learning rate | 5e-4 |
| Discount ($\gamma$) | 0.99 |
| GAE lambda ($\lambda$) | 0.95 |
| Init noise std | 1.0 |
| Desired kl | 0.008 |
| Ent-coef | 0 |

temporal information. The temporal dynamics are modeled through a 3-layer transformer encoder (dmodel=512, nheads=2), where each layer contains: Multi-head self-attention for capturing frame dependencies; Position-wise feed-forward network; Residual connections and layer normalization. Following the transformer encoder, we employ global average pooling across the temporal dimension and process the features through two residual blocks for enhanced representation learning. The final action predictor consists of a carefully designed MLP with progressively decreasing layer widths (256 → 128 units), each followed by ReLU activation, layer normalization, and dropout (p=0.1). The network outputs 8 consecutive action frames (dimension: 12 per frame) through a tanh-activated linear layer, ensuring actions remain within valid bounds.

We supervise the output action $\mathbf{a}_{\text{pred}}$ with negative log product loss with L2 regularization:

$$\mathcal{L}(\mathbf{a}_{\text{pred}}, \mathbf{a}_{\text{gt}}) = -\sum_{i=1}^{N} \log\left(1 - |\mathbf{a}_{\text{pred}}^{(i)} - \mathbf{a}_{\text{gt}}^{(i)}|1\right) + \lambda||\mathbf{a}_{\text{pred}}||_2^2 \tag{7}$$

## H.2 DIGITAL TWIN.

Table 11: Results for the real-world experiments

|       | box-w1 | box-w2 | box-w3 | bag-w1 | container | handbag |
|-------|--------|--------|--------|--------|-----------|---------|
| Wall  | 6/10   | 8/10   | 7/10   | 9/10   | 9/10      | 9/10    |
|       | box-e1 | box-e2 | bag-e1 | bag-e2 | plate     | handbag |
| Edge  | 10/10  | 7/10   | 10/10  | 8/10   | 5/10      | 9/10    |

Before leveraging the teacher-student distillation, we achieve zero-shot sim-to-real transfer by implementing a digital twin framework that bridges our simulation policy with the real dexterous arm-hand system. The framework operates through two parallel threads that enable real-time asynchronous communication between the simulation and real-world environments. The real-world thread continuously collects observations, including arm-hand proprioception and object pose information. Meanwhile, the simulation thread processes these observations to generate control actions, executes them in simulation and uses the resulting joint angles as target joint angles for PD control in the real robot system. The simulation environment is continuously synchronized with real-world by updating robot joint angles and object poses from real-world observations. Through the constant synchronization of simulation and real-world, we evaluate our RL polices following the similar setup as mentioned in Section 5.4. The results in Table 11 shows that our digital twin framework achieves robust performance on real-world objects, with success rates exceeding 80% for most objects.

