# OpenReview forum: "Dexterous Non-Prehensile Manipulation for Ungraspable Objects via Extrinsic Dexterity"
_ICLR.cc/2026/Conference — ICLR 2026 Conference Withdrawn Submission_

### Official Review · Reviewer_BRrQ · 2025-10-28

**Soundness:** 3
**Presentation:** 3
**Contribution:** 3
**Rating:** 6
**Confidence:** 4

**Summary:**

The paper introduces an approach that grasps ungraspable objects via extrinsic dexterity. The core idea is to leverage the environment, including the table edge and the wall, to aid grasping. The framework is composed of a high-level planner, a low-level controller, and a joint fine-tuning framework. The planner takes the environment observation as input and outputs the commands for the low-level policy. The low-level policy is composed of three primitive skills, including pushing, pivoting to the wall, and grasping the object from the table edge. The authors conducted extensive experiments to validate the effectiveness of the method.

**Strengths:**

- Good motivation. The paper is well motivated. Leveraging extrinsic dexterity to aid the process of grasping ungraspable objects is well-motivated. Endowing the policy with the ability to perceive the environment and utilizing the environment to aid the task completion is an interesting research direction.
- Well-designed method. The whole framework that incorporates the high-level planner and the low-level controller to solve the task is well-designed.
- Well-written and easy to follow.

**Weaknesses:**

- Limited technical contribution. Utilizing extrinsic dexterity to grasp ungraspable objects is not a new thing. Leveraging a hierarchical framework to solve the problem is also not a new concept. The design of the planner and the design of the low-level policies also lack originality and novelty.
- Restricted task complexity. For some objects, it seems that grasping with the edge or the well is not a necessity. For instance, to grasp the object shown in Figure 3, the robot only needs to grasp one side of the object. In some cases, pivoting the object using the table and then grasping the object is enough.

**Questions:**

- How do you judge whether the training is successful in the curriculum learning design?
- Why BC for distillation? Why not use DAgger?
- Could the method be utilized in more challenging scenarios, like grasping very thin objects?

---

### Official Review · Reviewer_CzW8 · 2025-10-31

**Soundness:** 3
**Presentation:** 3
**Contribution:** 2
**Rating:** 4
**Confidence:** 5

**Summary:**

This paper presents a sim-to-real approach for learning to manipulate objects through extrinsic contacts. To address the challenges of long-horizon execution and complex spatial interactions, the authors propose a hierarchical method that trains a coarse high-level planner and low-level primitive policies in simulation to decompose the manipulation task. The experiments focus on two scenarios: object pivoting against a wall and slide-and-grasp of flat objects on a tabletop.

**Strengths:**

1. The paper is clearly written and well-structured.

2. The studied problem—object manipulation via extrinsic contacts—is both interesting and relevant to advancing robotic dexterity.

3. The hierarchical decomposition of planning and control presents a reasonable direction for addressing long-horizon manipulation tasks.

**Weaknesses:**

1. The task decomposition appears to be too domain-specific. For example, training a binary classifier to decide whether to use the wall or the table edge restricts the method’s applicability and limits generalization across unseen environments.

2. The experimental results are not entirely convincing. See below.

(A) In the wall-pivoting experiments, the authors use only soft objects, which oversimplifies the problem. Prior work on object pivoting typically employs rigid objects [1], requiring precise force control and offering a more meaningful evaluation of the RL-based policy. In the current setting, the advantages of learning-based control are unclear, since teleoperation or scripted control could likely achieve similar results.

(B) The edge-slide-and-grasp experiment also appears too simple. The demonstrated behavior—pushing and grasping large, compliant objects—could potentially be achieved through motion planning with heuristic stopping conditions, rather than requiring learned policies.

3. The framework is not novel [2]. However, system papers are often less novel at a high level, so this is acceptable.

4. The terminology could be more precise. For instance, describing the objects as ungraspable is inaccurate—they are conditionally graspable, depending on their configuration. The naming of ExDex as a "framework" seems somewhat overstated, as the demonstrated skills remain relatively simple. It would strengthen the paper to showcase a more sophisticated manipulation behavior—e.g., grasping a flat object by leveraging the table as a pivot point to rotate and lift it.

[1] Xu et al. Rotating objects via in-hand pivoting using vision, force and touch. In IROS 2023.
[2] Zeng et al. Learning synergies between pushing and grasping with self-supervised deep reinforcement learning. IROS 2018.

**Questions:**

I wonder if the authors have tried to use imitation learning or model-based methods. I think the paper can be strengthened significantly by including more challenging setups as stated above.

---

### Official Review · Reviewer_hNeG · 2025-10-31

**Soundness:** 2
**Presentation:** 2
**Contribution:** 2
**Rating:** 4
**Confidence:** 4

**Summary:**

This paper introduces ExDex, a dexterous arm-hand system that learns to manipulate objects too large to grasp directly. The key idea is to exploit extrinsic dexterity — using environmental features like walls and table edges — to enable grasping through contact-rich, non-prehensile actions. The method combines a hierarchical reinforcement learning framework with a high-level planner (to select external contact goals) and a low-level controller (to execute pushing and grasping skills). The authors demonstrate both simulation and real-world transfer, reporting strong performance on diverse household objects and zero-shot generalization to unseen items.

**Strengths:**

**1. Real-world experiments add value.** The paper showcases real-world deployments of the system.

**Weaknesses:**

**1. Under-justified use of dexterity.** The paper titles itself with "Dexterous Non-prehensile Manipulation." I, however, do not see that experiments designed in the study are championing the use of dexterity. Even though a multi-fingered robot hand was employed, fingers are never truly used to the fullest extent, let alone the definition of dexterity can extend to the use of whole body surfaces and not limited to just the use of fingers. The poses of fingers in this study remain mostly unchanged throughout the trajectory.

**Therefore, on a problem level, I do not see a special contribution to a novel experimental setting for robot learning.** To me, this is not different from prior works done by Matt Mason and others on studying non-prehensile manipulation with simpler manipulators and grippers. I don't see why the objects are ungraspable; most of them can be grasped by a suction cup or standard grippers.

**2. The method is not addressing a general enough problem** The method proposed in the manuscript conjectures hand-crafted components such as pi_{wall} and pi_{edge}. This can work for a couple of tasks demonstrated in the study, but it will become intractable to implement given the infinite complexity of the physical world.

Overall, the proposed method seems to me like a complicated pipeline tailored for this specific toy task. But the toy task is a human-proposed artifact. There is no strict formal definition of what we mean by "non-prehensile manipulation." Therefore, I am very worried about what insights from this work can contribute to the coming age of robot learning in the long run.

This is not to mention that there are three stages in the proposed approach. It is unclear how this can scale to a large set of environments where there do not exist walls or edges.

**Questions:**

Please see the weaknesses section, thank you!

---

### Official Review · Reviewer_b7AY · 2025-11-10

**Soundness:** 2
**Presentation:** 3
**Contribution:** 2
**Rating:** 4
**Confidence:** 4

**Summary:**

This paper tackles the challenge of manipulating large, flat, or wide objects that cannot be grasped directly by a dexterous robotic hand. The authors propose ExDex, a hierarchical reinforcement learning framework that enables non-prehensile manipulation by leveraging environmental features such as table edges and walls. The system includes a high-level planner that predicts suitable environmental contact points and a low-level policy that executes dexterous pushing and grasping motions. They are then jointly fine-tuned to align sub-policies for smoother skill transitions. The framework is trained in simulation using domain randomization and transferred to a real-world UR5e + Inspire Hand setup via policy distillation. Experiments demonstrate successful object relocation and grasping on various shapes and configurations, achieving over 80% success rate on real-world trials.

**Strengths:**

(+) The paper extends the concept of extrinsic dexterity to multi-finger hands. This is a meaningful contribution that bridges environmental affordance exploitation with dexterous control.

(+) Separating high-level contact planning (wall vs. edge) and low-level control (pushing and grasping) leads to interpretable and trainable sub-policies. The joint fine-tuning stage is a thoughtful addition to address state mismatch between phases.

(+) The paper clearly explains training setups, policy structures, and the staged reward mechanism. The figures and diagrams effectively illustrate how each component contributes to the overall pipeline.

(+) The authors demonstrate successful real-world deployment.

**Weaknesses:**

(-) Limited diversity and scale of evaluation. Although 21 simulations and 10 real-world objects are reported, many are similar box-like shapes. The claim of “generalization to diverse objects” feels overstated;

(-) Slow motions. The result videos show relatively slow, quasi-static behaviors. It’s unclear if this limitation arises from the control policy, reward design, or simulation dynamics.

(-) Weak baseline selection and unclear fairness. The “Random Target” baseline unexpectedly achieves non-trivial success rates (50-60%) in Table 1, which suggests that the environment setup or success metric might be forgiving. More rigorous baselines, such as model-predictive control with explicit contact reasoning, would strengthen the evaluation.

(-) Limited analysis of sim-to-real challenges. The paper states a “successful zero-shot transfer,” but does not isolate what makes transfer hard: whether the bottleneck lies in perception (pose estimation), contact dynamics, or actuation latency.

(-) Real-world validation remains qualitative. The real experiments are convincing as proofs of concept, but lack comparative baselines or failure analysis under diverse conditions (e.g., clutter, lighting, or camera noise).

**Questions:**

What is the primary bottleneck for motion speed? Is it because of policy frequency?

Why does the Random Target baseline perform reasonably well (success rate)? Does this indicate that the planner’s learned predictions add only marginal benefit?

How does the method scale to environments with multiple contact affordances or cluttered scenes?

What are the dominant failure modes in real-world deployment? Is that pose estimation error, grasp instability, or object dynamics mismatch?

---

### Note · Authors · 2025-11-14

I have read and agree with the venue's withdrawal policy on behalf of myself and my co-authors.